# The E3 ligase Ubr3 regulates Usher syndrome and *MYH9* disorder proteins in the auditory organs of *Drosophila* and mammals

Tongchao Li[1], Nikolaos Giagtzoglou[2,3,4], Daniel F Eberl[5], Sonal Nagarkar Jaiswal[3,6], Tiantian Cai[7], Dorothea Godt[8], Andrew K Groves[1,3,7]*, Hugo J Bellen[1,2,3,6,7]*

[1]Program in Developmental Biology, Baylor College of Medicine, Houston, United States; [2]Jan and Dan Duncan Neurological Research Institute, Texas Children's Hospital, Houston, United States; [3]Department of Molecular and Human Genetics, Baylor College of Medicine, Houston, United States; [4]Department of Neurology, Baylor College of Medicine, Houston, United States; [5]Department of Biology, University of Iowa, Iowa City, United States; [6]Howard Hughes Medical Institute, Baylor College of Medicine, Houston, United States; [7]Department of Neuroscience, Baylor College of Medicine, Houston, United States; [8]Department of Cell and Systems Biology, University of Toronto, Toronto, Canada

**Abstract** Myosins play essential roles in the development and function of auditory organs and multiple myosin genes are associated with hereditary forms of deafness. Using a forward genetic screen in *Drosophila*, we identified an E3 ligase, Ubr3, as an essential gene for auditory organ development. Ubr3 negatively regulates the mono-ubiquitination of non-muscle Myosin II, a protein associated with hearing loss in humans. The mono-ubiquitination of Myosin II promotes its physical interaction with Myosin VIIa, a protein responsible for Usher syndrome type IB. We show that *ubr3* mutants phenocopy pathogenic variants of Myosin II and that Ubr3 interacts genetically and physically with three Usher syndrome proteins. The interactions between Myosin VIIa and Myosin IIa are conserved in the mammalian cochlea and in human retinal pigment epithelium cells. Our work reveals a novel mechanism that regulates protein complexes affected in two forms of syndromic deafness and suggests a molecular function for Myosin IIa in auditory organs.

*For correspondence: akgroves@bcm.edu (AKG); hbellen@bcm.edu (HJB)

## Introduction

Mechanosensory receptor cells have organelles derived from modified cilia or microvilli that contain protein complexes dedicated to the detection of, and adaptation to, mechanical force. Myosins, a family of eukaryotic actin-dependent motor proteins, play key roles in the assembly and function of mechanosensory protein complexes. In humans, pathogenic variants of six different myosin genes cause syndromic and non-syndromic deafness, and in many cases these myosins regulate either the assembly of the mechanotransduction apparatus of sensory hair cells, or constitute an integral part of the mechanotransduction complex itself (*Petit and Richardson, 2009*). For example, Myosin VIIa is a motor protein present in the tips of hair cell stereocilia where mechanotransduction occurs but it is also present in the cuticular plate that is important for the growth and stability of the stereociliary hair bundle (*Ahmed et al., 2013*). Pathogenic variants of MYO7A, the human homologue of *myosin VIIa*, can cause Usher syndrome, the leading cause of deaf-blindness (*Bonnet and El-Amraoui,*

**eLife digest** Over half of the world's population has hearing loss by the age of 65, and inherited forms of deafness are responsible for many of the hearing impairments in newborn children. Because the auditory organs that enable insects and mammals to hear work in similar ways, we have learnt a lot about genetic forms of deafness from identifying faulty genes in humans and mice, and studying their effects in model organisms.

Sensory cells in the inner ear respond to sound by detecting vibrations in the air and converting them into electrical impulses. A family of motor proteins called myosins play key roles in this conversion process. Mutations in the gene that produces one of these proteins, called myosin VIIa, cause an inherited deaf-blind disorder called Usher syndrome. Mutations in the gene for another type of myosin protein, called myosin II, also cause disorders associated with hearing loss, but it is not clear how they produce such effects.

Li et al. have used *Drosophila* fruit flies to explore the role of myosin proteins in hearing by looking for genes that prevent the insect's auditory organ from developing or working properly. The search identified one gene called E3 ubiquitin ligase (*ubr3*), which is required for the auditory organ to develop normally and had not previously been implicated in deafness. Mutating the *ubr3* gene caused a defect similar to that seen for mutations in the gene that produces the fruit fly equivalent of myosin VIIa.

Through genetic and biochemical studies, Li et al. found that in the fruit flies, myosin VIIa interacts with myosin II. This interaction is regulated by a chemical modification of myosin II that is controlled by *ubr3*. Li et al. showed that the equivalent mammalian proteins display the same behaviour in the cells of mammals. Therefore, mutations that affect myosin II alter how the protein interacts with myosin VIIa, which explains why myosin II is associated with deafness in humans.

In addition, Li et al. found that three other proteins that have been shown to cause Usher syndrome in humans have equivalents in flies and play a role in fly hearing. This will allow the *Drosophila* auditory organ to be further developed as a model system for future studies of deafness genes, and should provide insights into how specific genes are required for proper hearing in mammals.

*2012*), as well as the non-syndromic forms of deafness DFNA11 (*Liu et al., 1997*) and DFNB2 (*Weil et al., 1997*). Dominant mutations in *MYH9*, which encodes Myosin IIa, cause a number of syndromes which are now grouped as 'MYH9-related disorders' (*Seri et al., 2003*). Many *MYH9*-related disorder patients exhibit sensorineural deafness, and variants of *MYH9* have also been reported in non-syndromic deafness DFNA17 (*Lalwani et al., 2000*). However, the cellular basis of deafness in pathogenic variants of *MYH9* is unclear as MYH9 is widely expressed within the inner ear (*Etournay et al., 2010*; *Lalwani et al., 2000*; *Meyer Zum Gottesberge and Hansen, 2014*; *Mhatre et al., 2006*).

One approach to identifying new genes that regulate the development and function of mechanosensory organs is to exploit the power of *Drosophila* to conduct forward genetic screens. The auditory organ of *Drosophila,* Johnston's organ, is localized in the second antennal segment. Johnston's organ responds to near-field sound, gravity and wind flow transduced by motion of the third antennal segment (*Boekhoff-Falk and Eberl, 2014*; *Gopfert and Robert, 2001*; *Kamikouchi et al., 2009*; *Yorozu et al., 2009*). Although the organs and cells that mediate hearing in vertebrates and *Drosophila* are morphologically different, they share a striking evolutionary conservation of molecular and functional properties (*Albert and Gopfert, 2015*; *Boekhoff-Falk and Eberl, 2014*). The transcriptional cascades that control key aspects of chordotonal development in flies and hair cell development in vertebrates are regulated by conserved transcription factors, such as the Atonal/Atoh1 family proteins (*Jarman et al., 1993*; *Wang et al., 2002*). In addition, myosins such as Myosin VIIa, encoded by the gene *crinkled* in *Drosophila*, that function in mammalian hair cell mechanotransduction, are also conserved in *Drosophila* and are required for hearing (*Todi et al., 2005b*, *2008*). Therefore, other molecular pathways and regulatory protein partners that function in hearing are also likely to be shared between insects and vertebrates.

Here, we describe a novel ubiquitination pathway in *Drosophila* that functions to regulate the activity and physical interaction of two proteins implicated in deafness, Myosin II and Myosin VIIa. We identified an E3 ubiquitin ligase, *ubr3*, from a collection of lethal mutations on the *Drosophila* X chromosome (*Haelterman et al., 2014*; *Yamamoto et al., 2014*), whose loss of function causes morphological defects in the Johnston's organ. Ubr3 negatively regulates the mono-ubiquitination of Myosin II and modulates Myosin II-Myosin VIIa interactions, which are required for normal development of Johnston's organ. We show that *ubr3* mutations are phenotypically similar to known pathogenic variants of Myosin II and that Ubr3 physically and genetically interacts with *Drosophila* homologues of the Usher syndrome proteins Protocadherin 15 (Pcdh15) and Sans. We also show that Myosin IIa interacts with Myosin VIIa in the mouse cochlea and human retinal pigment epithelial cells. Our study reveals a novel conserved ubiquitination pathway in the auditory organs of flies and mammals.

## Results

### A forward genetic screen identifies Ubr3, an E3 ligase necessary for correct morphological development of the *Drosophila* auditory organ

Johnston's organ is a large chordotonal organ located in the second antennal segment of *Drosophila* (*Figure 1A*). Each organ consists of more than 200 functional units or scolopidia (*Kamikouchi et al., 2006*), containing 2~3 sensory neurons, each bearing a single specialized mechanosensitive cilium (*Figure 1B–B′*) (*Eberl and Boekhoff-Falk, 2007*). The neuronal cilium is enveloped by a tube-like scolopale cell, which forms septate junctions with a cap cell that attaches the scolopidium to the cuticle of the third antennal segment. A ligament cell attaches the other end of the scolopidium to the cuticle of the second antennal segment (*Figure 1A,B*). Each scolopidium is thus suspended between the second and third antennal segments, and rotation of the third antennal segment leads to flexion of the scolopidia and stimulation of the sensory neurons (*Boekhoff-Falk and Eberl, 2014*).

To identify genes required for auditory organ development and function, we screened a collection of X-chromosome induced lethal mutations (*Haelterman et al., 2014*; *Yamamoto et al., 2014*). We generated mutant clones in Johnston's organ through FLP/FRT-mediated mitotic recombination using an *eyeless-FLP* driver. We assessed morphological defects in the constituent cell types of the scolopidia by co-labeling with cell type-specific markers (neurons: ELAV and HRP; scolopale cells and scolopale space: Prospero and Eyes shut; actin bundles in scolopale cells: phalloidin; ligament cells: Repo; *Figure 1B–B′* and *Figure 1—figure supplement 1A*) and identified seven complementation groups that showed a range of different morphological defects in Johnston's organ. One complementation group, *ubr3* (*Zanet et al., 2015*), exhibits a specific detachment of the scolopidia from the third antennal segment of mutant clones (arrows in *Figure 1C,D*). Extracellular electrophysiological recordings in flies with *ubr3* mutant clones in Johnston's organ showed significantly reduced auditory transduction (*Figure 1E* and *Figure 1—figure supplement 1B*). The incomplete reduction in sound-evoked potentials was due to the small size of mutant clones in Johnston's organ (*Figure 1C and D*).

*ubr3* encodes a 2219 amino acid protein homologous to the mammalian RING-type E3 ubiquitin ligase n-recognin 3 (*UBR3*) (*Figure 1F*) (*Huang et al., 2014*; *Meisenberg et al., 2012*; *Tasaki et al., 2007*; *Yang et al., 2014*; *Zanet et al., 2015*; *Zhao et al., 2015*). Ubr3 contains a UBR substrate binding domain, a RING E3 ligase domain and a C-terminal auto-inhibitory (AI) domain. To determine the expression pattern and protein localization of Ubr3, we stained antennae at 50% of pupariation, when the scolopidia mature, with a specific Ubr3 antibody (*Zanet et al., 2015*) (*Figure 1F*). Ubr3 is broadly expressed in the second and third segments of the antenna (*Figure 1—figure supplement 1C*), containing Johnston's organ and olfactory neurons respectively. A prior study reported expression of mouse *UBR3* in multiple sensory tissues, including the inner ear and olfactory epithelium (*Tasaki et al., 2007*). In Johnston's organ, Ubr3 is enriched in the apical tips of neurons and scolopale cells (*Figure 1G–I*, arrowheads, and *Figure 1—figure supplement 1D*).

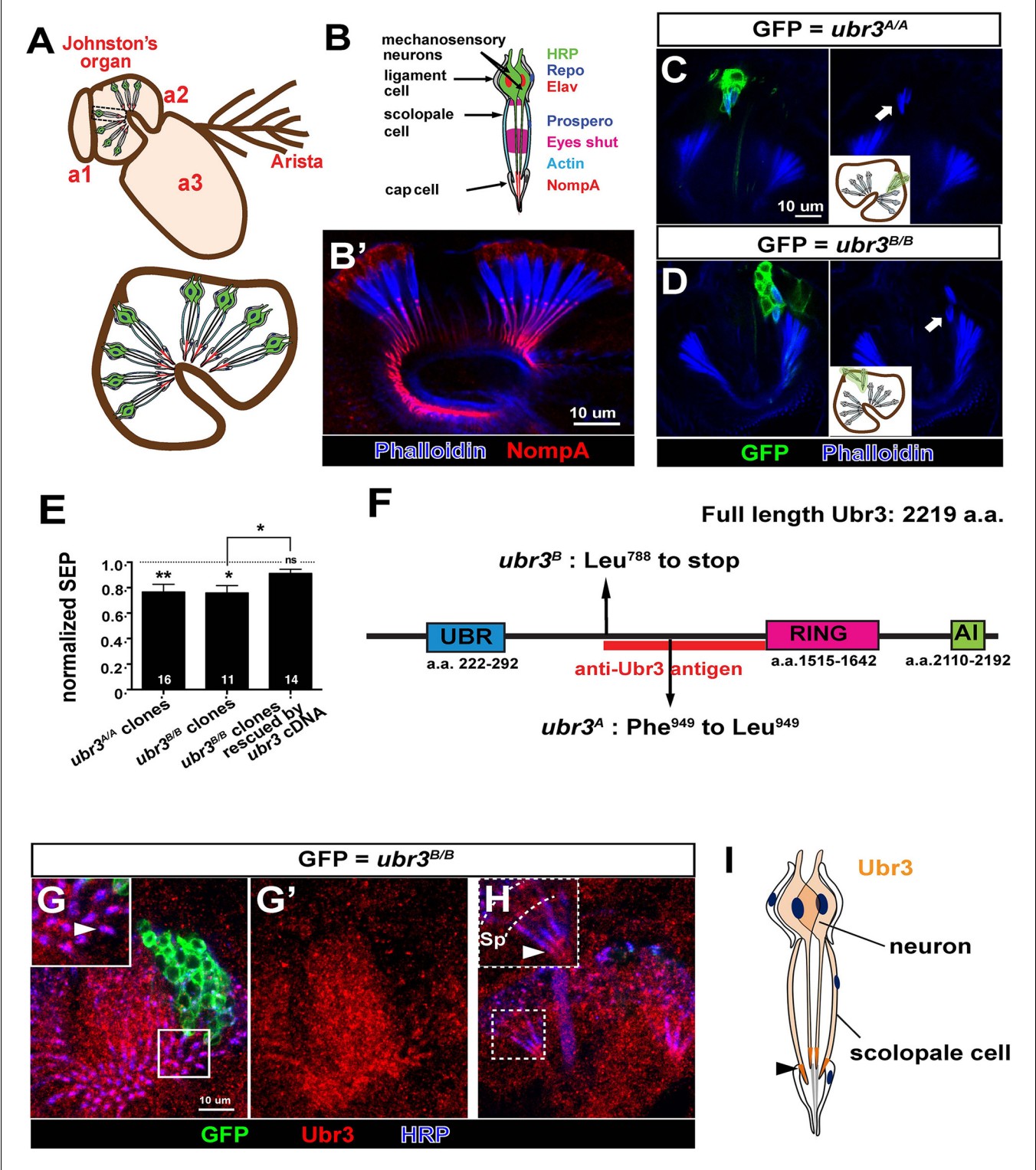

**Figure 1.** *ubr3* regulates auditory organ development in *Drosophila*. (A) The structure of the *Drosophila* auditory organ, Johnston's organ. The tips of the neuronal cilia are anchored to the cuticle of the third antennal segment by a dendritic cap containing an extracellular glycoprotein, NompA (B) A single scolopidium (corresponding to the box in A) shows the markers used to label various structures and cells in the scolopidium. (B') Immunolabeling of Johnston's organ with NompA (red) and phalloidin (actin, blue). (C–D) Pupal Johnston's organs bearing *ubr3* mutant clones were stained with phalloidin to label the actin bundles of scolopale cells. Some *ubr3* mutant cells (labeled by GFP) exhibit scolopidia detached from the apical junction of Johnston's organ (arrows). (E) Extracellular electrophysiological recordings in flies bearing *ubr3* mutant clones in Johnston's organ (left 2 columns) and

*Figure 1 continued on next page*

*Figure 1 continued*

in flies bearing *ubr3* cDNA rescued mutant clones (right column). The data are normalized to flies heterozygous for the corresponding mutations. Numbers of flies recorded are shown in the columns. Error bars show SEM. Statistical symbols show results of *t*-tests with Welch's correction as needed (*p<0.05; **p<0.01; ns, not significant). (**F**) Schematic diagram showing the conserved domains of the Ubr3 protein and the molecular lesions (Phe$^{949}$ > Leu and Leu$^{788}$ > STOP) identified in the *ubr3$^{A}$* and *ubr3$^{B}$* alleles respectively. The red bar shows the epitope used to generate anti-Ubr3 antibody. (**G–G'**) A single confocal cross-section shows co-immunolabeling of Johnston's organ with anti-Ubr3 (red) and HRP (neurons, in blue). *ubr3$^{B/B}$* mutant clones were generated and labeled with GFP. Ubr3 protein is localized to neuronal cilia marked by an arrowhead. (**H**) A longitudinal section of Johnston's organ labeled by anti-Ubr3 (red) and HRP (blue). Ubr3 localizes not only to neuronal cell bodies but faint expression is also seen in cilia. Arrowhead labels enriched Ubr3 proteins in apical ciliary tips. Scolopale cell bodies (Sp) are outlined by dashed lines. (**I**) Diagram shows distribution of Ubr3 proteins in Johnston's organ, including its enrichments in the apical tips of the neuronal cilia and the scolopale cells (arrowhead).

The following figure supplement is available for figure 1:

**Figure supplement 1.** Ubr3 is required for normal development of Johnston's organ.

## *ubr3* genetically interacts with *myosin VIIa* in the *Drosophila* auditory organ

Loss of *ubr3* in Johnston's organ leads to detachment of scolopidia from the hinge of the second and third antennal segment (*Figure 1C,D*). This phenotype has previously been reported for only one other *Drosophila* gene, *crinkled* (also known as *myosin VIIa* or *myoVIIa*) (*Todi et al., 2005b*, *2008*) (*Figure 2A*). The detachment of scolopidia in *myoVIIa* mutants is accompanied by a severely altered distribution of a glycoprotein, NompA, that links the tip of the neuronal cilium with the antennal cuticle (*Figure 2A–C*) (*Todi et al., 2005b*, *2008*). NompA is the homologue of vertebrate tectorins, a glycoprotein family present in the tectorial membrane of the cochlea (*Chung et al., 2001*). We observed a similar change of NompA distribution in *ubr3* mutant cells (*Figure 2D,E*), which can be rescued by over-expressing wild type Ubr3 proteins with an *actin-Gal4* driver (*Figure 2—figure supplement 1A–B*). However, over-expression of an E3 enzymatic inactive form of Ubr3 (*Li et al., 2016*) did not rescue the detachment of scolopidia in *ubr3$^{B/B}$* mutant cells (*Figure 2—figure supplement 1A–B*), suggesting that Ubr3 regulates apical attachment of scolopidia through its E3 ligase activity. To confirm if the ubiquitination function of Ubr3 is necessary for its role in Johnston's organ, we knocked down *ubcD6* using RNAi. *ubcD6* encodes the E2 enzyme that interacts with Ubr3 in mammals (*Tasaki et al., 2007*; *Zanet et al., 2015*). As shown in *Figure 2F*, knock down of *ubcD6* also causes scolopidial detachment (*Figure 2F*), suggesting that the phenotype in *ubr3* mutant cells is caused by failure of ubiquitination of one or more target proteins.

MyoVIIa is an unconventional myosin expressed in hair cells in the vertebrate inner ear and has been shown to be localized to the tip of the stereocilia close to the proposed sites of mechanotransduction (*Grati and Kachar, 2011*). In *Drosophila*, we found that MyoVIIa is abundant in the scolopale cells of Johnston's organ (*Figure 2G,H* and *Figure 2—figure supplement 1C,D–D'*), being especially enriched at their apical tips (arrows in *Figure 2—figure supplement 1D–D'*). The specific and unique phenotype observed in *ubr3* and *myoVIIa* mutant cells suggests that Ubr3 and MyoVIIa may function in the same genetic pathway. To test if *ubr3* interacts genetically with *myoVIIa*, we first generated *ubr3$^{B/B}$; myoVIIa$^{RNAi}$* cells. However, we observed nearly 100% detachment of scolopidia (data not shown), similar to that seen with the *myoVIIa$^{RNAi}$* knockdown alone (*Figure 2A*). We then over-expressed MyoVIIa in *ubr3* mutant cells and observed a strong enhancement of the *ubr3* mutant phenotype (*Figure 2I*) whereas over-expression of MyoVIIa in wild-type cells did not cause any detachment of scolopidia (*Figure 2I*). These observations indicate a specific genetic interaction between *ubr3* and *myoVIIa*.

## Ubr3 negatively regulates the mono-ubiquitination of MyoII and MyoVIIa-MyoII interaction

Our data suggest that ubiquitination by Ubr3 regulates the apical attachment of scolopidia in the *Drosophila* hearing organ, and the similarity of *ubr3* and *myoVIIa* mutants raises the possibility that it may regulate the abundance, localization or function of MyoVIIa. However, mutant clones of *ubr3* in Johnston's organ do not exhibit altered protein levels or subcellular localization of MyoVIIa (*Figure 2I*), suggesting that Ubr3 instead regulates MyoVIIa function. To determine if Ubr3 regulates

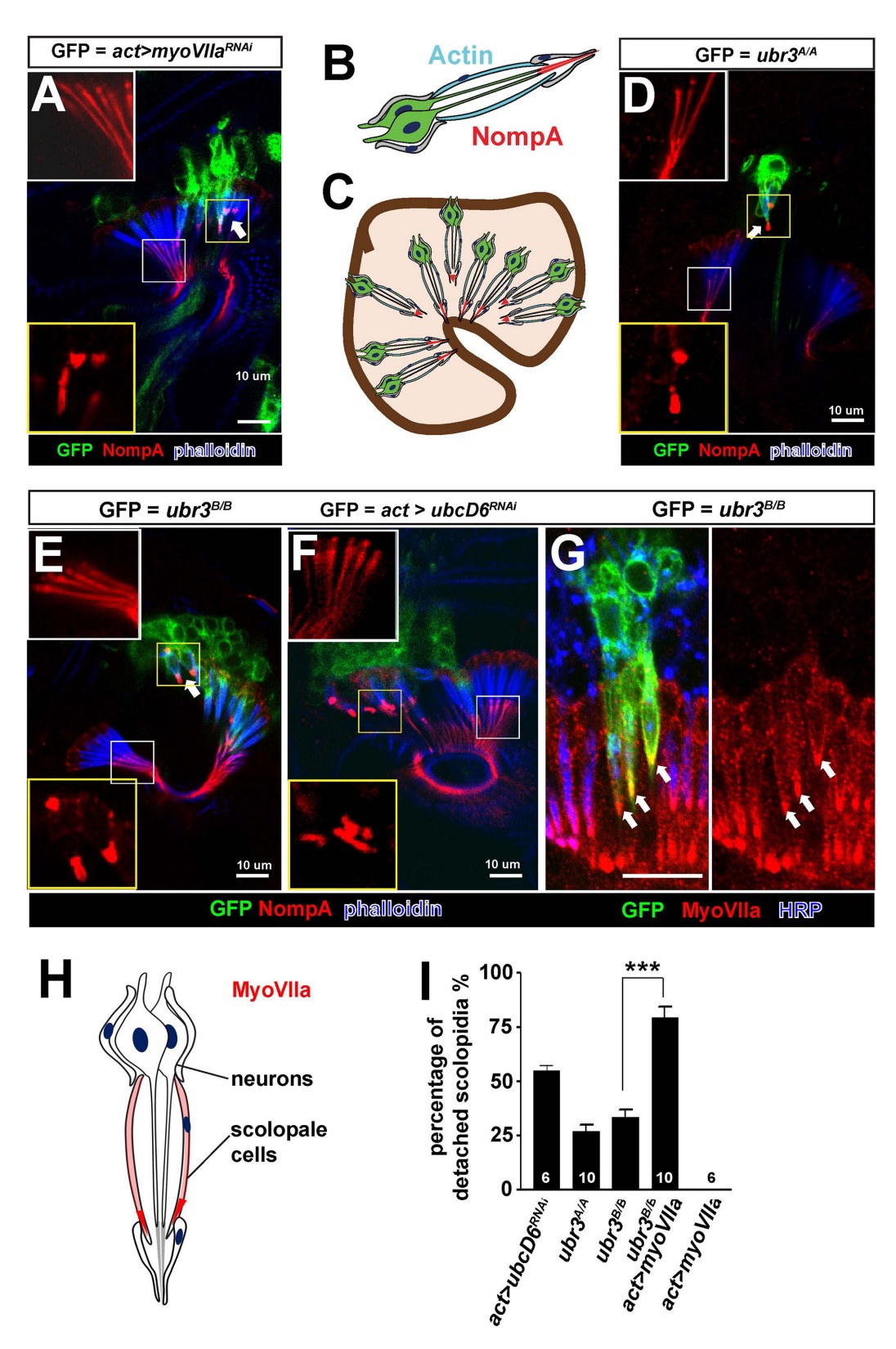

**Figure 2.** Ubr3 genetically interacts with MyoVIIa. (**A**) The normal filamentous structure of NompA in the apical junction of wild-type cells (white box) collapses into puncta in the detached scolopidia in flies in which *myoVIIa* is knocked down (yellow box). Arrow indicates detached scolopidia. (**B**) A
*Figure 2 continued on next page*

Figure 2 continued

diagram shows actin (cyan) and NompA (red) in a single scolopidium. (C) A diagram illustrates the detachment of scolopidia and altered NompA pattern. (D–E) The normal filamentous structure of NompA in the apical junction of wild-type cells (white boxes) collapses into puncta in the detached scolopidia in *ubr3* mutant cells (labeled by GFP) (yellow boxes). (F) The normal filamentous structure of NompA in the apical junction of wild-type cells (white box) collapses into puncta in the detached scolopidia in cells over-expressing *ubcD6* RNAi construct (labeled by GFP) (yellow box). (G) Immunolabeling of Johnston's organ with *ubr3* mutant clones (marked by GFP, green) by anti-HRP (neurons, blue) and anti-MyoVIIa antibody (red). Arrows indicate detached *ubr3* mutant scolopidia. (H) A diagram shows localization of MyoVIIa (red) in neuronal cilia and scolopale cells. (I) Quantification of detached scolopidia in the $ubr3^{A/A}$ and $ubr3^{B/B}$ clones, *ubcD6* RNAi clones, and wild type or mutant clones over-expressing *myoVIIa*. Error bars show SEM. Numbers of flies quantified are shown in the columns. (***p<0.001)

The following figure supplement is available for figure 2:

**Figure supplement 1.** *ubr3* mutants phenocopy *myoVIIa* mutants.

ubiquitination of MyoVIIa, we purified GFP-MyoVIIa fusion protein from wild type or *ubr3* mutant clone cells in eye-antennal discs in third instar larvae followed by western blotting (*Figure 3A,B*, arrow). We then assessed the ubiquitination of MyoVIIa with an anti-poly- and mono-ubiquitin antibody or an anti-poly-ubiquitin antibody (*Figure 3B*). Interestingly, although we did not observe ubiquitination of MyoVIIa, we detected mono-ubiquitination of a MyoVIIa-interacting protein, which migrates lower than MyoVIIa-GFP as a 220 kDa protein (arrowhead in *Figure 3B*). The mono-ubiquitination of this 220 kDa protein is increased in *ubr3* mutant cells (*Figure 3B,C*), suggesting that Ubr3 indirectly regulates the function of MyoVIIa by ubiquitinating an unknown, interacting partner.

To identify this unknown protein, we performed mass spectrometry from the 220 kDa band and found the *Drosophila* homologue of the heavy chain of non-muscle Myosin II (MyoII), encoded by the gene *zipper*. To verify that MyoII is the target of Ubr3, we probed the membrane with an anti-MyoII antibody. The MyoII antibody detected a band at the same molecular weight as the ubiquitinated proteins (*Figure 3B*, shown by anti-MyoII antibody). Interestingly, we observed more MyoII co-precipitating with MyoVIIa in *ubr3* mutant cells compared to wild-type cells (*Figure 3B*), suggesting that loss of *ubr3* leads to a stronger MyoVIIa-MyoII interaction.

## MyoII regulates apical attachment of scolopidia in Johnston's organ

To test if MyoII has a similar function as MyoVIIa in Johnston's organ, we generated a *myoII-GFP-myoII* knock-in line by integrating an artificial exon that encodes GFP with flexible linkers into an intron of *myoII* (*Nagarkar-Jaiswal et al., 2015a*, *2015b*; *Neumüller et al., 2012*; *Venken et al., 2011*). This GFP-tagged protein is fully functional, as it complements a deletion spanning the *myoII* gene. We detected abundant MyoII protein in both neurons and scolopale cells (*Figure 3D,E*). We observed an enrichment of MyoII proteins at the apical tips of cilia, in the vicinity of apically-enriched MyoVIIa protein (*Figure 3D'–D'',E*). In addition, over-expression of MyoII in *ubr3* mutant cells increased the penetrance of detached scolopidia from 30% to 60% (*Figure 3G,H*), while over-expression of MyoII in wild-type cells did not affect scolopidial structure (*Figure 3F,H*). Thus, our data show that MyoII and MyoVIIa interact with *ubr3*, and that over-expression of either MyoII or MyoVIIa in Johnston's organ can cause scolopidial detachment provided *ubr3* is also mutated.

To understand the function of mono-ubiquitination of MyoII, we fused a ubiquitin coding sequence to the carboxyl terminal of MyoII cDNA (MyoII-Ub) to artificially mimic the mono-ubiquitinated MyoII (*Figure 3I*). When we over-expressed this MyoII-Ub construct in clones in Johnston's organ, we again observed detached scolopidia and an altered NompA pattern (*Figure 3J–K'*). The penetrance of the phenotype is lower than that observed in *ubr3* mutant cells (~5%), probably because carboxyl terminal fused ubiquitin does not function as optimally as those at the normal ubiquitination sites in MyoII.

## Ubr3 regulates the mono-ubiquitination of MyoII through Skp1-Cullin1-F-box-protein (SCF) E3 ubiquitin ligases

We were surprised to observe that loss of the E3 ubiquitin ligase *ubr3* caused increased mono-ubiquitination of MyoII (*Figure 3B,C*). One potential mechanism is that Ubr3 may negatively regulate a second ubiquitin ligase complex that in turn mono-ubiquitinates Myo II. We tested the expression of

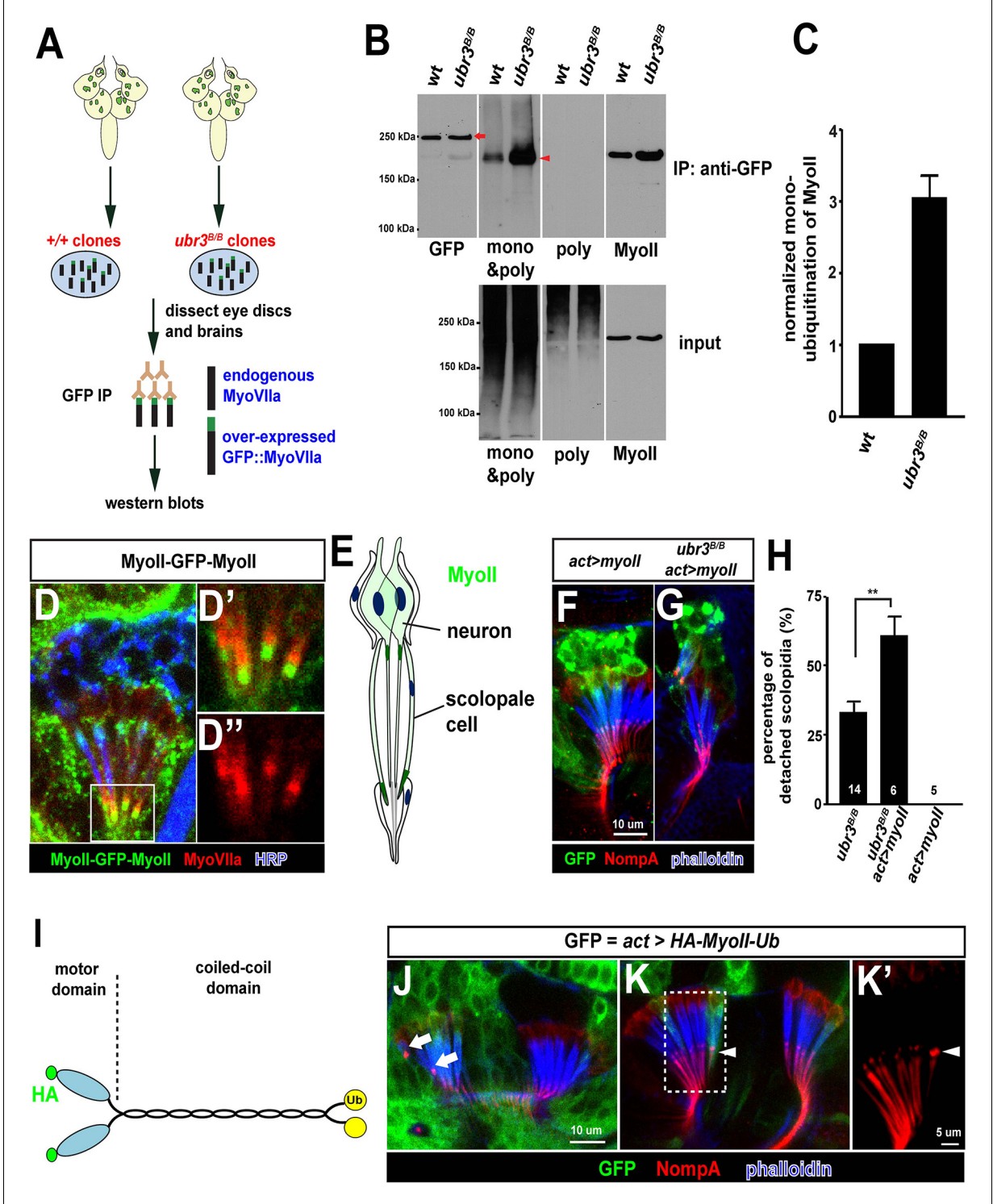

**Figure 3.** Ubr3 negatively regulates the mono-ubiquitination of MyoII and MyoII-MyoVIIa interaction. (**A**) A GFP amino terminal tagged MyoVIIa construct, GFP-MyoVIIa, is expressed in wild type (control) or *ubr3* mutant clones in larval eye-antennal discs. The lysate of eye-antenna discs and brains containing *ubr3* mutant cells expressing GFP-MyoVIIa protein was immunoprecipitated with GFP nanobody-conjugated beads and examined on western blots. (**B**) Western blots with anti-GFP, anti-poly & mono-ubiquitin, anti-mono-ubiquitin and anti-MyoII antibodies. (**C**) Quantification of mono-ubiquitination of MyoII normalized by total amount of MyoII proteins from (**B**). (**D–D''**) Immunolabeling of GFP (green), MyoVIIa (red) and HRP (neurons, in blue) in Johnston's Organ from a *myoII-GFP-myoII* transgenic fly. **D'** and **D''** show magnified images of the region shown by white box in **D**. (**E**) Distribution of MyoII proteins in Johnston's organ. (**F**, **G**) Wild type MyoII over-expressing cells show normal apical structures of scolopidia, whereas

*Figure 3 continued on next page*

*Figure 3 continued*

$ubr3^{B/B}$ mutant cells expressing wild type MyoII exhibit enhanced detachment of scolopidia. (H) Quantification of detached scolopidia in $ubr3^{B/B}$ mutant cells, $ubr3^{B/B}$ mutant cells over-expressing MyoII and wild type cells over-expressing MyoII. Error bars show SEM. Numbers of flies quantified are shown in the columns. (I) Diagram shows HA-MyoII-Ub in which MyoII is fused to a Ub coding sequence on the carboxyl terminal. (J–K') Johnston's organ containing HA-MyoII-Ub expressing clones (labeled by GFP, green) is immunolabeled by anti-NompA (red) and phalloidin (actin, in blue). Arrow marks detached scolopidia. (K–K') One MyoII-Ub expressing scolopidium exhibits accumulated NompA at the tips, but stays attached to the cuticle from the third segment (arrowheads). This may be a defective scolopidium just before detaching, suggesting that NompA mis-localization happens prior to apical detachment, as opposed to being a consequence of detachment.

The following figure supplement is available for figure 3:

**Figure supplement 1.** Ubr3 regulates MyoVIIa through Cul1.

widely expressed E3 ligase, Cullin1, a component of the SCF E3 ubiquitin ligase complex (*Deshaies, 1999*; *Wu et al., 2005*), and found that it is strongly up-regulated in *ubr3* mutant clones in imaginal discs. We observed a similar up-regulation of Cul1 in *ubr3* mutant cells in Johnston's organ (*Figure 3—figure supplement 1A*), suggesting that Ubr3 negatively regulates Cul1. To test if Cul1 was also implicated in scolopidial attachment, we over-expressed Cul1 in wild-type cells and found that it recapitulated the apical detachment of scolopidia that was seen in *ubr3* and *myoVIIa* mutant clones (*Figure 3—figure supplement 1B,F*). This suggests that increased Cul1 expression is likely to cause scolopidial detachment in *ubr3* mutant clones. In addition, RNAi-mediated down-regulation of Cul1 produced a similar detachment of scolopidia (*Figure 3—figure supplement 1C,F*). The observation that both gain- and loss-of-function of Cul1 produce the same specific detachment phenotype in Johnston's organ implies that a critical range of Cul1 activity or level is necessary for its proper function. To test if other components of the SCF E3 ubiquitin ligase complex, SkpA and Roc1a (*Murphy, 2003*; *Noureddine et al., 2002*), affect scolopidial attachment we performed RNAi experiments and also observed detachment (*Figure 3—figure supplement 1D–F*). These data indicate that the phenotypes associated with Cul1 over-expression and down-regulation are mediated by the SCF E3 ligase complex. However, the mono-ubiquitination of MyoII is increased in *skpA* mutant cells, (*Figure 3—figure supplement 1G–H*), again showing that the SCF is not the direct E3 ligase that mono-ubiquitinates MyoII (*Figure 3—figure supplement 1I*). Together, our data show that Ubr3 negatively regulates Cul1 (SCF) E3 ligase, and that both E3 ligases control the mono-ubiquitination of MyoII and apical attachment of scolopidia.

## Pathogenic variants of *MYH9* cause similar defects as *ubr3* mutations in Johnston's organ

Dominant pathogenic variants in *MYH9*, one of three human paralogues of *myoII*, cause *MYH9*-related disease in human (*Ma and Adelstein, 2014*; *Seri et al., 2003*) and affect hearing to varying degrees (*Pecci et al., 2014*; *Verver et al., 2015*). We over-expressed four mutant forms of *Drosophila myoII* in Johnston's organ that contain variants commonly found in individuals with *MYH9*-related disorders who have sensorineural hearing loss (*Franke et al., 2007*) (*Figure 4—figure supplement 1A*) and observed scolopidial detachment with variable penetrance in all variant forms of *myoII* (*Figure 4A–E*). In addition, we tested whether these variants alter the localization of MyoII. Over-expression of all four MyoII variants led to the formation of puncta in neurons, in contrast to a diffuse localization of wild type MyoII (*Figure 4F–J*). We observed similar puncta formation when the constitutively ubiquitinated form of MyoII, MyoII-Ub, was over-expressed in neurons (*Figure 4K*). However, the MyoII variants and MyoII-Ub exhibited comparable localization with wild type MyoII when expressed in the scolopale cells (*Figure 4F'–J'*). In addition, we performed immunoprecipitation of GFP-tagged wild type or variants of MyoII expressed in eye-antenna discs using *ey-Gal4*. We found that $MyoII^{D1847K}$ exhibits an increased interaction with MyoVIIa (*Figure 4—figure supplement 1B*). The data for the other three are either not changed or there is a decrease in their interaction. This suggests that a mis-regulated MyoII-MyoVIIa interaction may be present in some patients with MYH9-related disorders. Our data suggest that pathogenic variants of MyoII exhibit dominant toxic effects in Johnston's organs and lead to similar phenotypes to those seen in *ubr3* mutants.

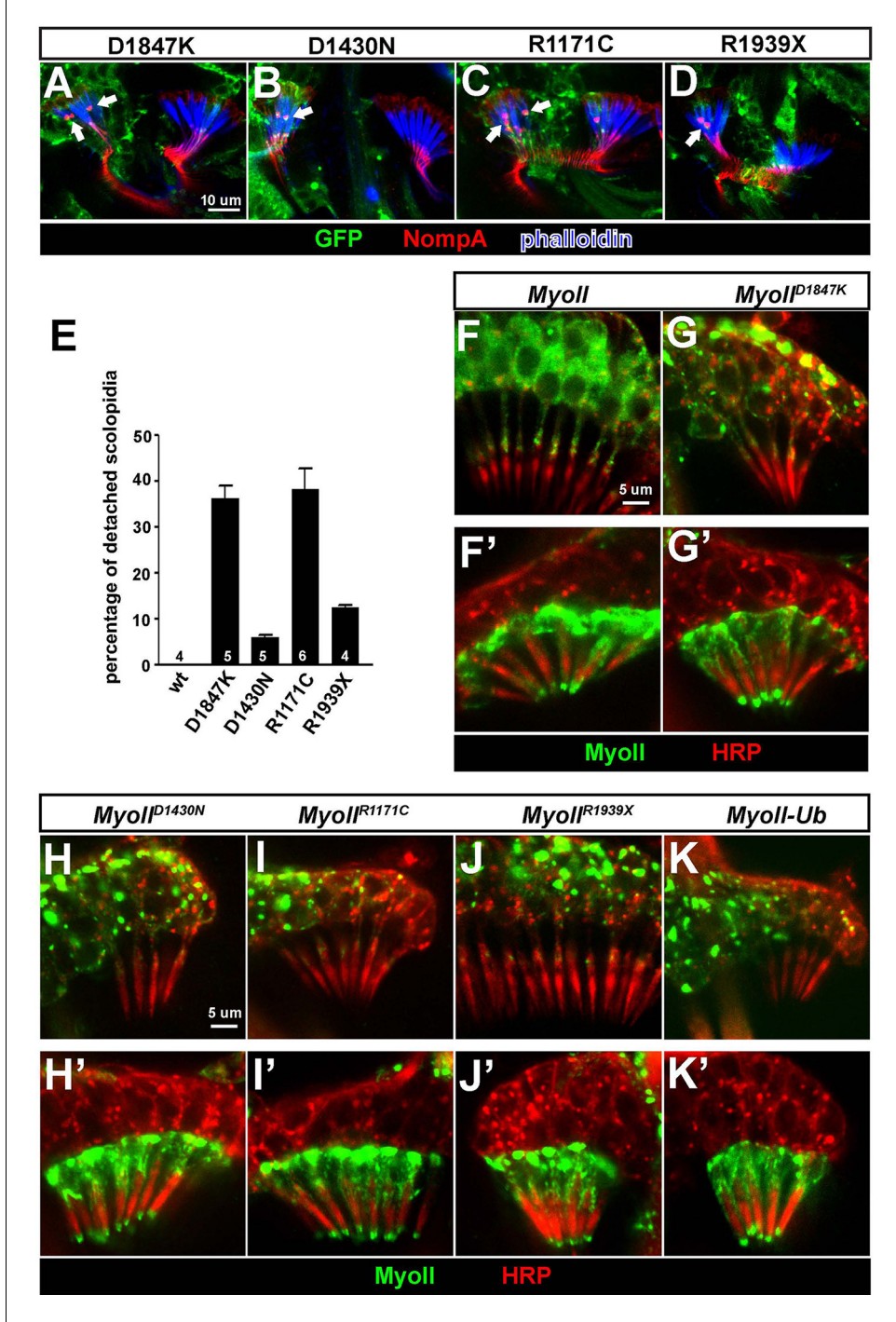

**Figure 4.** Over-expression of pathogenic variants of MyoII in Johnston's organ leads to similar defects as *ubr3* mutants. (**A–D**) Johnston's organs over-expressing four different GFP tagged MyoII[mut] in clones (labeled with GFP in green) are immunolabeled with anti-NompA (red) and phalloidin (actin, in blue). Arrows mark detached scolopidia. (**E**) Quantification of detached scolopidia in the clone cells expressing the four MyoII mutant forms shown in **A**. Error bars show SEM. Numbers of flies quantified are shown in the columns. (**F–K'**) Johnston's organs expressing GFP-MyoII, MyoII mutant forms or MyoII-Ub in neurons using *nsyb-Gal4* driver (**F–K**) or in scolopale cells using *nompA-Gal4* (**F'–K'**) are immunolabeled with anti-GFP (green) and HRP (neurons, in red).

The following figure supplement is available for figure 4:

**Figure supplement 1.** Pathogenic mutations of MyoII.

## Mammalian UBR3 regulates MyoIIa (*MYH9*) and its interaction with MyoVIIa

To examine if mammalian UBR3 regulates MyoIIa (encoded by *MYH9*) and its interaction with Myo-VIIa, we turned to a human retinal pigment epithelial cell line ARPE-19, one of the few cell lines that express MyoVIIa (*Soni et al., 2005*) as well as MyoIIa and Ubr3 (see below). In wild type ARPE-19 cells, MyoIIa predominantly localizes to stress fiber-like structures (*Figure 5A,B*, arrows). A small pro-portion of MyoIIa is present in puncta in the cytoplasm (arrowheads). Interestingly, most MyoVIIa protein in ARPE-19 cells co-localize with MyoIIa in both stress fibers and cytoplasmic puncta, although some MyoVIIa does not overlap with MyoIIa in the peri-nuclear region (empty arrowhead).

To test whether UBR3 regulates *MYH9* in ARPE-19 cells, we knocked down UBR3 with three inde-pendent siRNAs, all of which down-regulated UBR3 mRNA to ~30% (*Figure 4—figure supplement 1C*). Upon down-regulation of UBR3, the ARPE-19 cells become elongated with long protrusions (*Figure 5A*). We observed a consistent decrease of stress fiber-localized MyoIIa and MyoVIIa and an increase of MyoIIa-MyoVIIa co-localized puncta in all siRNA transfections (*Figure 5B–D*). To test whether these changes are caused by defective MyoIIa, we treated the cells with different doses of blebbistatin, an inhibitor for MyoIIa (*Bond et al., 2013*). Interestingly, cells treated with low doses of blebbistatin (2–4 µM in *Figure 5E,F*) mimic *UBR3* knock-down cells (*Figure 5A,B*). The stress fibers are decreased in the UBR3 knocked-down cells or in the cells treated with blebbistatin (*Figure 5—figure supplement 1A*), suggesting that the formation of stress fibers in these cells are mis-regu-lated. These results imply that loss of UBR3 leads to defects in MyoIIa function.

To further examine the MyoIIa interaction with MyoVIIa in mammals, we examined the localization of MyoIIa protein in the neonatal mouse cochlea. MyoIIa protein is expressed weakly in both hair cells and supporting cells (*Figure 5G–H*). In the cell body of hair cells, MyoIIa localizes at the apical surface in neonatal mice (*Etournay et al., 2010*) and faintly at junctions with supporting cells (*Ebrahim et al., 2013*) (*Figure 5—figure supplement 1B*, arrowheads). However, MyoIIa is restricted to hair cell stereocilia in six day old mice (*Figure 5I–I'*, arrowheads). To assess whether MyoIIa physically interacts with MyoVIIa, we performed an immunoprecipitation assay from inner ear lysates of five day old mice. Similar to what we observed with *Drosophila* MyoII and MyoVII, MyoVIIa co-immunoprecipitated MyoIIa (*Figure 5J*). In addition, we detected a ubiquitinated form of MyoIIa in ARPE-19 cells (*Figure 5K*). Therefore, the MyoIIa-MyoVIIa ineraction and ubiquitination of MyoIIa are conserved in mammals.

## Homologues of Usher syndrome type 1 genes are expressed in the *Drosophila* auditory organ and interact genetically and physically with Ubr3

Previous studies have shown that at least five members of the USH1 protein family, including Myo-VIIa, can interact to form a complex (*Adato et al., 2005*; *Boeda et al., 2002*; *Kazmierczak et al., 2007*; *Reiners et al., 2005*; *Senften et al., 2006*) and localize to the tips of hair cell stereocilia in mice (*Figure 6A*) (*Grati and Kachar, 2011*; *Hilgert et al., 2009*). These protein complexes are thought to interact with and regulate hair cell mechanotransduction channels (*Gillespie and Muller, 2009*). Although *Drosophila* homologues of several USH proteins have been identified (*D'Alterio et al., 2005*; *Demontis and Dahmann, 2009*; *Todi et al., 2005b*), only MyoVIIa has been shown to be expressed in the scolopidia of Johnston's organ and to be required in *Drosophila* for hearing (*Todi et al., 2005b*, *2008*). To test whether other components of the Usher protein complex are conserved in the fly hearing organ and whether Ubr3 can interact with other *Drosophila* USH proteins in addition to MyoVIIa, we characterized the expression, phenotypes and protein interac-tions of several USH1 homologues.

The *Drosophila* homologue of the vertebrate Usher syndrome gene *PCDH15* is *Cad99C* (*D'Alterio et al., 2005*; *Schlichting et al., 2006*). To examine expression of *Cad99C*, we integrated an artificial exon that encodes GFP with flexible linkers into an intron of *Cad99C* using the MiMIC technique (*Neumüller et al., 2012*; *Venken et al., 2011*). Cad99C was enriched in the apical part of the actin-rich scolopale cells, in a similar manner to MyoVIIa and MyoII (*Figure 6B,C*). To examine if Ubr3, Cul1 or MyoVIIa physically interact with Cad99C, we performed co-immunoprecipitation assays in S2 cells. *Cad99C* encodes a transmembrane cell adhesion protein with 11 cadherin repeats in the extracellular domain and a short intracellular domain (*Figure 6—figure supplement 1A*).

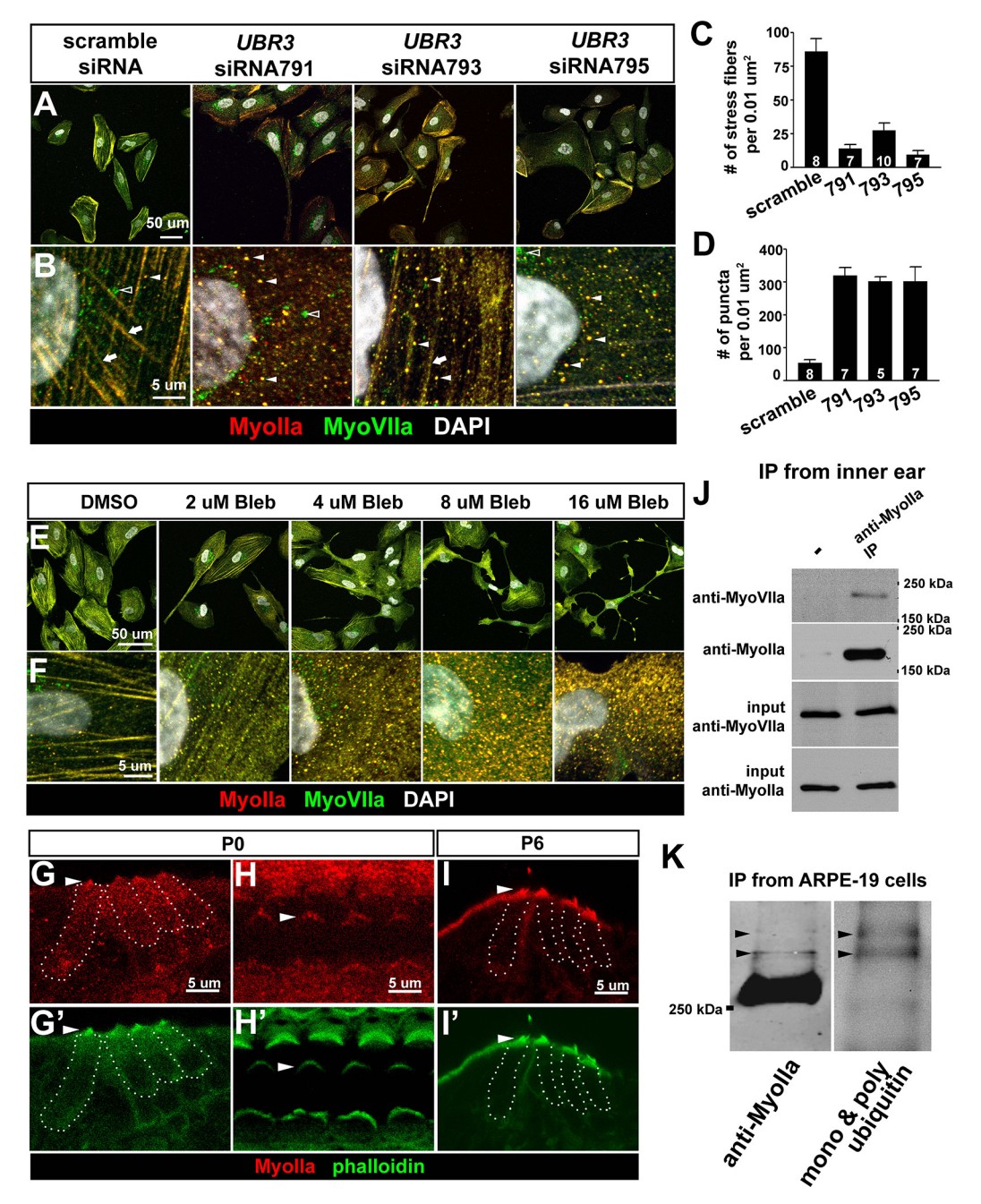

**Figure 5.** The function of Ubr3 is conserved in vertebrate cells. (**A–B**) Cultured ARPE-19 human cells transfected with indicated siRNAs are co-immunolabeled by anti-MyoIIa and anti-MyoVIIa antibodies and DAPI (white). Arrows: stress fibers. Arrowheads: MyoIIa-MyoVIIa co-localized puncta. Empty arrowheads: MyoVIIa positive, MyoIIa negative puncta. (**C–D**) Quantifications of stress fiber number and MyoIIa-MyoVIIa puncta shown in **A–B**. Error bars show SEM. Numbers of cells quantified are shown in the columns. (**E, F**) ARPE-19 cells treated with indicated concentrations of blebbistatin for 30 min followed by immunolabeling with anti-MyoIIa (red) and anti-MyoVIIa (green) antibodies and DAPI. Low concentration (2–4 μM) of blebbistatin treatment resulted in elongated ARPE-19 cells with protrusions, similar as *UBR3* siRNA-treated cells. Further increasing the dosage of blebbistatin (8–16 μM) resulted in cells with a more branched, tree-like morphology. The number of puncta correlated with the concentration of blebbistatin, suggesting a specific change in MyoII-MyoVIIa interactions. (**G–G'**) Immunolabeling of a cochlear section from a neonatal mouse with anti-MyoIIa (red) and phalloidin (green). Hair cells are outlined by dashed lines. (**H–H'**) Surface view of whole mount cochlea from a neonatal mouse immunolabeled with anti-MyoIIa (red) and phalloidin (green). Arrowheads mark V-shaped stereocilia (labeled by phalloidin, green). (**I–I'**) Immunolabeling of cochlear section from P6 pup with anti-MyoIIa (red) and phalloidin (green). Arrowheads mark stereocilia (shown by phalloidin staining in green). (**J**) Co-immunoprecipitation with anti-

*Figure 5 continued on next page*

*Figure 5 continued*

MyoIIa antibody from P5 cochlear lysate followed by western blotting. (**K**) MyoIIa was purified from ARPE-19 cells through immuno-precipitation, followed by western blot. Arrowheads indicate ubiquitinated MyoIIa (shown by FK2 antibody).

The following figure supplement is available for figure 5:

**Figure supplement 1.** MyoII proteins localize close to cell membrane in the hair cells of mouse cochlea.

Interestingly, the 37 kDa intracellular fragments of Cad99C (arrowheads) co-precipitated with Myo-VIIa (*Figure 6E*, empty arrowhead), Cul1 (square) and UBR (empty square), whereas the full length Cad99C protein (arrow) did not co-precipitate. To verify the specific interaction with the cleaved, intracellular fragments of Cad99C, we performed the reciprocal co-immunoprecipitation assay (*Figure 6—figure supplement 1B*) and observed consistent results. This indicates that only the short carboxyl-terminal fragments of Cad99C bind to Ubr3. These results are consistent with previous finding that Cad99C interacts with MyoVIIa through carboxyl-terminal domain in *Drosophila* ovary (*Glowinski et al., 2014a*). We also tested whether a second Usher syndrome homologue, Sans, interacts with Ubr3 and Cul1. As shown in *Figure 6E*, MyoVIIa, Cul1, and the UBR domain all interact with Sans. Altogether, these results demonstrate that Ubr3 and Cul1 physically interact with Cad99C and Sans.

To test if the altered MyoVIIa-MyoII interactions seen in *ubr3* mutants also affect the function of *Drosophila* USH protein homologues, we examined mutants for *sans* and *Cad99C* in the presence or absence of *ubr3* mutations. *sans^245^*, a null mutant for *sans* (*Demontis and Dahmann, 2009*), does not display detachment of scolopidia (*Figure 6G*). However, *Cad99C^57A^*, a null allele (*Schlichting et al., 2006*), exhibits detached scolopidia, albeit at a much lower frequency (1%) than *ubr3* or *myoVIIa* mutants (*Figure 6D,G*). However, removal of one copy of *Cad99C* or *sans* in *ubr3* mutant clones increased the penetrance of scolopidial detachment from 30% to 90% or 60%, respectively (*Figure 6F–G*). These data show that several *Drosophila* homologues of USH1 proteins cooperate in the Johnston's organ and suggest that *ubr3* genetically interacts with and may regulate a number of Usher complex proteins. This suggests that altering the strength of interactions between MyoII and MyoVIIa also affects the function of other USH1 proteins.

## Discussion

Functional and molecular homologies between hearing organs of *Drosophila* and mammals have allowed the use of *Drosophila* to identify new genes involved in hearing (*Boekhoff-Falk and Eberl, 2014*; *Senthilan et al., 2012*). Using a forward genetic screen based on the phenotype associated with the loss of MyoVIIa (*Todi et al., 2005a, 2008*), we identified a set of proteins that affect ubiquitination and regulate the function of MyoVIIa. We discovered that *ubr3* mutations increase the mono-ubiquitination of MyoII and the association of MyoII and MyoVIIa. This increased association causes defects consistent with a reduction of MyoVIIa function, namely the detachment of scolopidia from the cuticle of Johnston's organ. Significantly, over-expressing either myosin in Johnston's organ has no effect unless *ubr3* is also mutated, suggesting that it is not the absolute levels of either myosin that are important for their function in Johnston's organ, but rather the level of their ubiquitin-dependent interaction.

Myosins are known to be regulated by phosphorylation through their regulatory light chain (*Ito et al., 2004*). However, far less is known about the post-translational regulation of myosin heavy chains. Ubiquitination as a regulator of myosin function has not yet been reported. When we knocked down *UBR3* in human ARPE-19 cells, we observed defects similar to inhibition of MyoII by blebbistatin, suggesting that a certain threshold of mono-ubiquitination is sufficient to attenuate the activity of MyoII. Thus, it is possible that phosphorylation and ubiquitination function respectively as an 'accelerator' and 'brake' for MyoII activity. Our data suggests that ubiquitination of MyoII is regulated by multiple E3 ligases (*Figure 7A–A'*), in which Ubr3 negatively regulates the Cul1-SCF complex directly or indirectly. This complex probably regulates an E3 ligase that directly ubiquitinates MyoII.

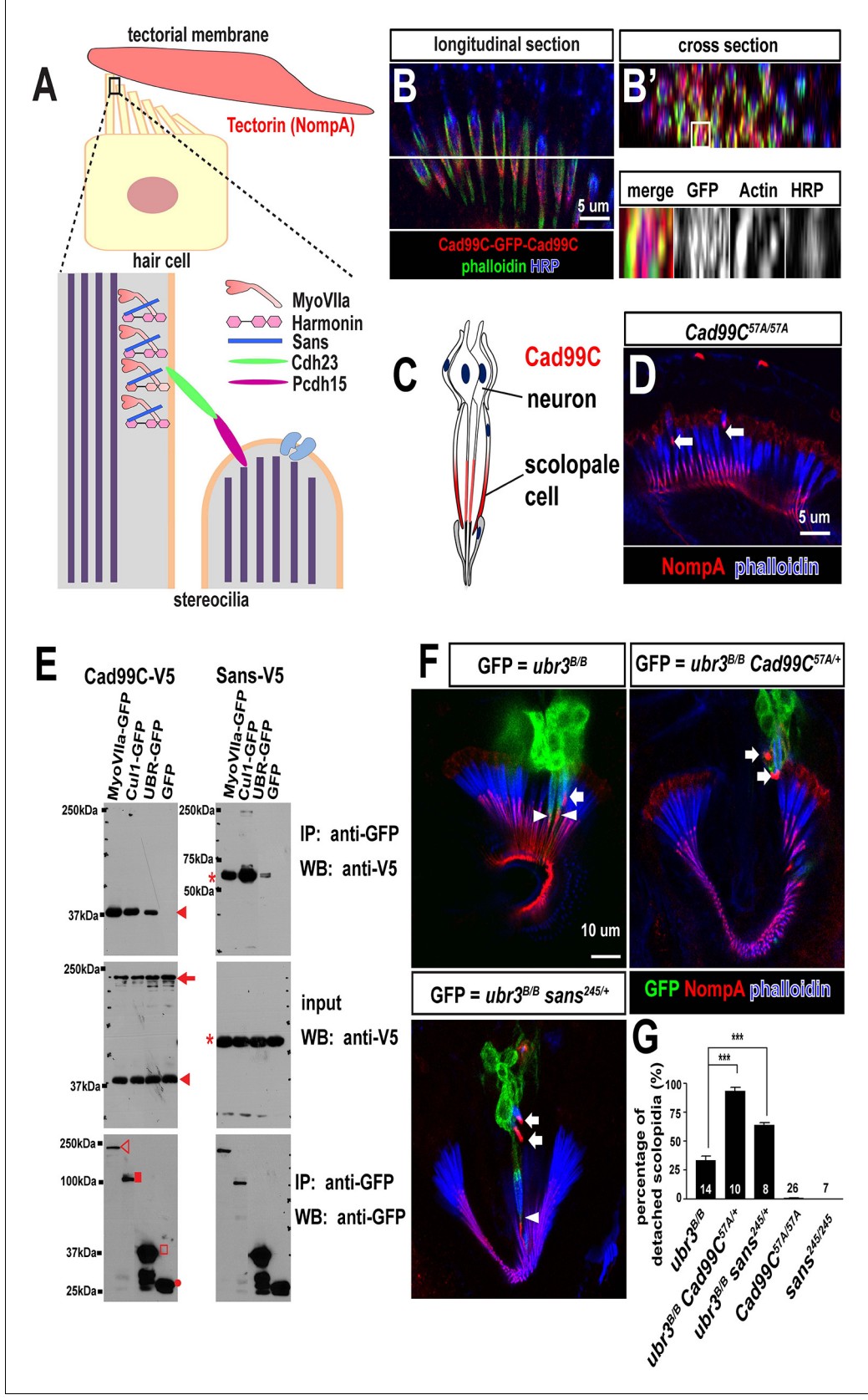

**Figure 6.** Ubr3, Cul1 and MyoVIIa interact with *Drosophila* homologues of Usher proteins. (**A**) Diagram of a vertebrate hair cell and localization of USH1 proteins in stereocilia. (**B–B′**) Johnston's organ of a fly carrying a

*Figure 6 continued*

homozygous GFP knock-in allele of *Cad99C* is labeled with HRP (blue, neurons), phalloidin (actin, in green, scolopale cells) and anti-GFP (red, Cad99C proteins). (C) Localization of Cad99C proteins in neuronal cilia and the tip region of scolopale cells. (D) Johnston's organ of *Cad99C$^{57A}$* mutant is stained with phalloidin (actin, blue) and NompA (red). Arrows indicate two detached scolopidia. (E) S2 cells transfected with the indicated constructs were lysed and immunoprecipitated with GFP nanobody conjugated beads. Western blots were performed with various antibodies. In the input fraction for the immunoprecipitation, two proteins can be detected: a short 37 kDa carboxyl terminal domain (arrow) and a full length 250 kDa protein (arrowhead). MyoVIIa-GFP (empty arrowhead), Cul1-GFP (square), UBR-GFP (empty square), and untagged GFP proteins (dot). UBR-GFP was used because we could not detect expression of Ubr3 full length protein here. (F) Johnston's organs containing *ubr3$^{B/B}$*, *ubr3$^{B/B}$ Cad99C$^{57A/+}$* or *ubr3$^{B/B}$ sans$^{245/+}$* clone cells (GFP, green) are stained with phalloidin (actin, blue) and NompA (red). Arrows mark detached GFP+ scolopidia and arrowheads mark un-detached GFP+ scolopidia. (G) Quantification of detached scolopidia. Error bars show SEM. Numbers of flies quantified are shown in the columns. ***p<0.001

The following figure supplement is available for figure 6:

**Figure supplement 1.** Ubr3, Cul1 and MyoVIIa interact with *Drosophila* homologues of Usher proteins.

---

We were surprised to find that MyoII ubiquitination appears to be regulated by a series of E3 ligases including Ubr3 and Cul1. Either these E3 ligases function in a sequential order, or alternatively, function collaboratively (*Metzger et al., 2013*). A previous study showed that Fbx2, an F-box protein that binds Skp1 in SCF E3 ligase complex, is specifically expressed in mouse cochlea, and that Fbx2 deficient mice exhibit selective cochlear degeneration (*Hartman et al., 2015*; *Nelson et al., 2007*). Our results argue that the SCF E3 ligase regulates auditory function through MyoIIa and MyoVIIa, two proteins associated with deafness and expressed in hair cells. In addition, multiple E3 ligases, including Ubr3 and a yet unknown E3 ligase that directly ubiquitinates MyoIIa, function in this pathway. It is interesting that Ubr3 and Cul1 can bind to a complex containing MyoVIIa, Cad99C and Sans in *Drosophila* S2 cells. It is possible that Ubr3, Cul1 and the unknown E3 ligase that mono-ubiquitinates MyoII all interact in a protein complex, and that the Usher proteins Cad99C or Sans may facilitate ubiquitination of MyoII or may be ubiquintinated themselves.

It is currently unclear whether MyoII and MyoVIIa interact directly or indirectly as part of a protein complex in Johnston's organ. Since both MyoII and MyoVIIa are localized in adjacent zones in the tips of scolopale cells in Johnston's organ (*Figure 7A*), it is possible that either myosin is required for the subcellular transport or localization of the MyoVIIa – MyoII complex. The precise mechanical function performed by these two myosins in the scolopale cells of Johnston's organ involves other homologues of Usher syndrome type I genes given that the morphological phenotype of *ubr3* mutants is enhanced by loss of either Sans or Cad99C (*Figure 7A,A''*). We have shown that homologues of three Usher syndrome genes interact together in *Drosophila* to cause a mechanical failure phenotype which appears to be orthologous to that seen in the mechanosensory hair cells of vertebrates (*Figure 7B*). Moreover, by revealing a physical interaction between MyoII and MyoVIIa in the hearing organs of insects and mammals and by recapitulating morphological defects in Johnston's organ by overexpressing *Drosophila* versions of known pathogenic human *MYH9* variants, our study offers a potential mechanism for the hearing deficits associated with *MYH9*-related disorders.

## Materials and methods

### Fly strains and genetics

*y w ubr3$^A$ FRT19A/FM7c Kr-Gal4, UAS-GFP* and *y w ubr3$^B$ FRT19A/FM7c, Kr-Gal4, UAS-GFP* flies (*Li et al., 2016*; *Zanet et al., 2015*) were crossed to *tub-Gal80, y w, eyFLP, FRT19A; Act-Gal4, UAS-CD8-GFP/CyO* to generate GFP-labeled *ubr3* homozygous mutant clones. All UAS transgenic flies were generated through φC31-mediated transgenesis (*Venken et al., 2006*). Additional strains used in the study are as follows: *Cad99C$^{57A}$* (*Schlichting et al., 2005*), *sans$^{245}$* (*Demontis and Dahmann, 2009*), *UAS-ubr3* (*Zanet et al., 2015*), *UAS-GFP-myoVIIa* (*Todi et al., 2005b*), *UAS-cul1/CyO* (*Ou et al., 2002*). *UAS-GFP-myoII*, *UAS-GFP-myoII$^{D1847K}$*, *UAS-GFP-myoII$^{D1430N}$*, *UAS-GFP-*

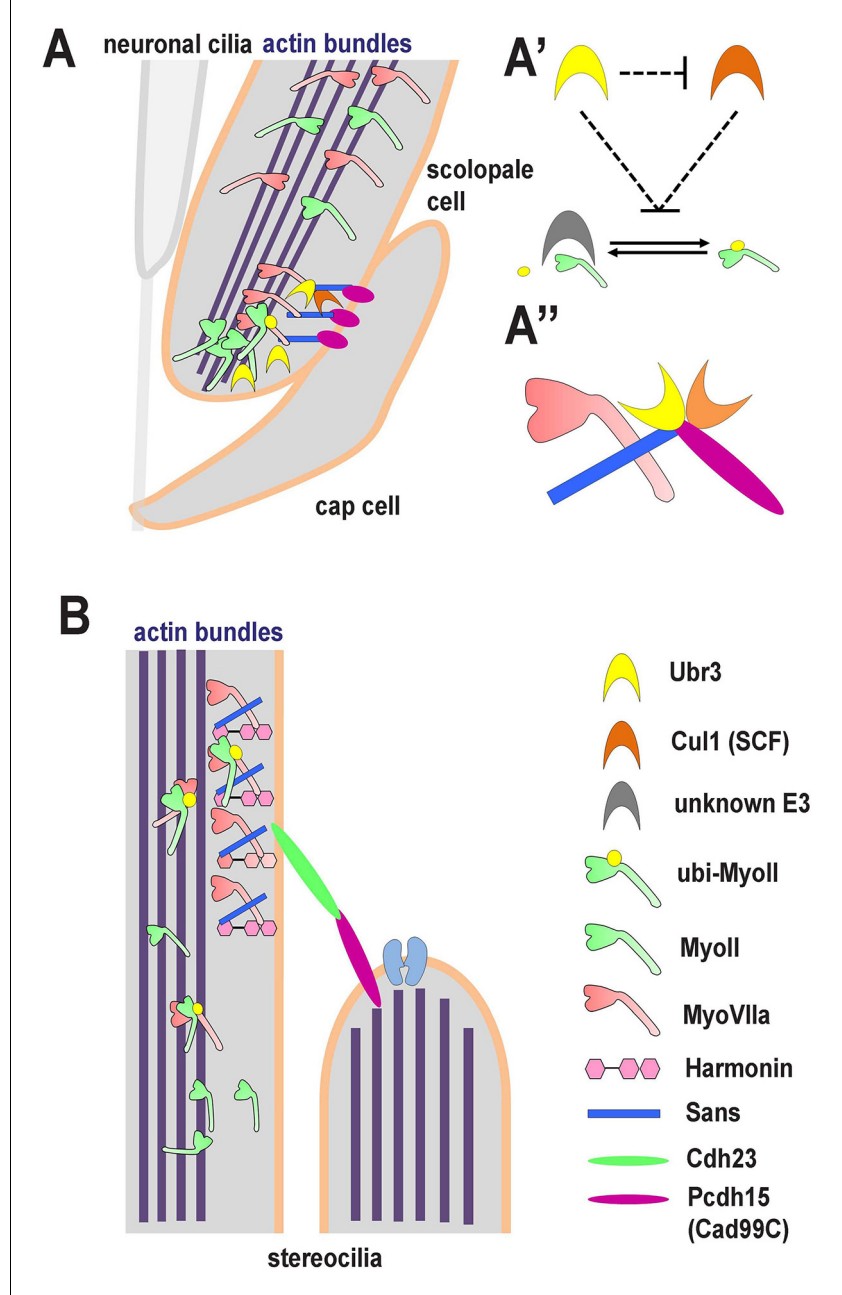

**Figure 7.** A novel ubiquitination pathway regulates MyoII-MyoVIIa interactions in the auditory sensory organs of *Drosophila* and mammals. (**A**) In *Drosophila*, MyoVIIa and MyoII are present in the apical regions of scolopidia of Johnston's organ and are enriched in the tips of the scolopale cells where they contact the cap cell. Ubiquitination of MyoII promotes its interaction with MyoVIIa, the precise level of which is crucial for anchoring the apical junction complexes of the scolopidia. It is possible that the motor activity of either myosin is necessary to transport the complex to the tips of the scolopale cell. Both MyoVIIa and MyoII likely bind to the actin bundles in the scolopale cells and regulate apical attachment of scolopidia. Two *Drosophila* homologues of Usher syndrome type I proteins, Cad99C (Pcdh15) and Sans, interact with MyoVIIa, Ubr3 and Cul1 in a protein complex. It is not clear whether Cad99C mediates attachment to the cap cells as a homodimer or as a heterodimer with another adhesion molecule. (**A'**) Ubr3 negatively regulates the level of Cul1 protein. Both Ubr3 and Cul1 inhibit ubiquitination of MyoII indirectly through a pathway involving a third unknown E3 ligase. (**A''**) MyoVIIa, Cad99C, Sans, Ubr3 and Cul1 interact as a protein complex. (**B**) In mammalian hair cells, an USH1 protein complex which includes MyoVIIa, Sans, Harmonin and Cadherin-23 is present close to the stereocilia tips. We speculate that MyoII interacts with

*Figure 7 continued on next page*

*Figure 7 continued*

MyoVIIa, and that this interaction is promoted by ubiquitination of MyoII. The motor activity of MyoII or MyoVIIa may be required for transport of the MyoVIIa-MyoII-USH1 protein complex to the stereocilia tips.

*myoII^{R1171C}*, *UAS-GFP-myoII^{R1939X}* are kind gifts from Dr. Kiehart (*Franke et al., 2007*). *UAS-myo-VIIa^{RNAi}* (P{GD1408}v9265), *UAS-roc1a^{RNAi}* (P{GD8596}) (*Dietzl et al., 2007*), *UAS-UbcD6^{RNAi}* (TRiP. HMS02466), *UAS-cul1^{RNAi}* (TRiP. HM05197), *UAS-skpA^{RNAi}* (*Ni et al., 2009*) (TRiP.HM05185) (Bloomington *Drosophila* Stock Center, Bloomington, IN), *nsyb-Gal4* (FlyBase ID: FBst0051635, generated by Dr. Julie Simpson, unpublished), *nompA-Gal4* (*Chung et al., 2001*). All flies were maintained at room temperature and crossed on standard food at 25°C.

## Immunolabeling and imaging

Fly tissues were dissected in PBS in room temperature and fixed with 3.7% formaldehyde in PBS for 20 min, followed by permeabilization with 0.2% Triton-X100 in PBS. For whole-mount mouse cochlear staining, cochleas from neonatal mice and 6-day old mice were dissected in PBS, with the spiral ganglia and Reissner's membrane removed to expose the organ of Corti. For sections of neonatal ear tissue, animal heads were fixed 1–2 hr in 4% PFA at room temperature, cryoprotected in 30% sucrose in PBS at 4°C, embedded in OCT compound, and cryosectioned at 14 μm. For sections of 6-day old mice cochlea, inner ears were dissected in PBS, fixed 1 hr in 4% PFA at room temperature, and decalcified in 0.5 M EDTA pH8.0 for three days at 4°C. Then the decalcified inner ears were cryoprotected in 30% sucrose in PBS at 4°C, embedded in OCT compound, and cryosectioned. The immunohistochemistry procedure followed standard protocols with some minor modifications. The primary antibodies and secondary fluorescently-labeled antibodies used in this paper were: chicken anti-GFP (1:1000, Abcam, United Kingdom), rat anti-ELAV (1:1000, 7E8A10, DSHB, Iowa City, IA) (*Robinow and White, 1991*), rabbit anti-HRP (1:1000, Jackson Immunoresearch Laboratories Inc., West Grove, PA), mouse anti-Pros (1:100, MR1A, DSHB) (*Spana and Doe, 1995*), mouse anti-Repo (1:100, 8D12, DSHB) (*Alfonso and Jones, 2002*), rabbit anti-NompA (1:250) (*Chung et al., 2001*), mouse anti-Futsch (1:100, 22C10, DSHB) (*Fujita et al., 1982*; *Zipursky et al., 1984*), guinea pig anti-Ubr3 (1:1000) (*Zanet et al., 2015*), rabbit anti-Cul1 (1:250) (*Wu et al., 2005*), guinea pig anti-MyoVIIa (GP6 1:2000, used in *Figure 2G* and *Figure 2—figure supplement 1C–D*) (*Glowinski et al., 2014b*), Mouse anti-MyoVIIa (1:10, 138–1, DSHB, used in *Figure 5A,B,E,F*), rabbit anti-MyoIIa (1:500, Gene Tex (Irvine, CA) GTX113236, used for ARPE-19 cell and mouse cochlear experiments shown in *Figure 5*), Alexa488- (1: 1000, Life Technologies, Carlsbad, CA), Cy3- and Cy5- conjugated affinity purified donkey secondary antibodies (1: 1000, Jackson ImmunoResearch Laboratories). Images were acquired using LSM510 and LSM710 confocal microscopes (Zeiss, Germany) and examined and processed using LSM viewer (Zeiss), ZEN (Zeiss) and Photoshop (Adobe) software.

## Cloning, plasmid constructs

To clone the pUASTattB-HA-MyoII-Ub construct, *myoII (zipper)* cDNA was amplified from genomic DNA of *UAS-GFP-myoII* transgenic flies. The human Ub sequence was cloned from a construct from Dr. Janghoo Lim from Yale University. *myoII* cDNA and human Ub sequences were then cloned into pUASTattB vector through EcoRI, NotI and KpnI. An HA sequence was inserted in the primer. To clone pUASTattB-Cad99C-V5 and pUASTattB-sans-V5 constructs, *Cad99C* and *sans* cDNAs were amplified from BDGP Gold cDNA clones LP14319 and LD20463 (*Drosophila* Genomics Resource Center, Bloomington, IN) and sub-cloned into pUASTattB vector through NotI and KpnI. A V5 sequence was inserted into the primer at the carboxy terminal of *Cad99C* or *sans* in frame. All constructs were verified through Sanger sequencing before use.

## Cell culture assays

S2 cells were cultured at 25°C in Schneider's medium (Life Technologies) plus 10% heat-inactivated fetal bovine serum (Sigma, St. Louis, MO), 100 U/mL penicillin (Life Technologies), and 100 μg/mL streptomycin (Life Technologies). Cells were split every 3 days and plated at a density of $10^6$ cells/

well in 12-well cell culture plates for experiments. Transfections were carried out using Effectene transfection reagent (Qiagen, Germany). Cells were harvested 48 hr after transfection for biochemical assays.

ARPE-19 cells were cultured in at 37°C in 5% CO2 in air in DMEM: F-12 medium (ATCC) with 10% fetal bovine serum, as described by the ATCC (http://www.atcc.org/). Cells were split when reaching 80–90% confluence. siRNAs were transfected using Lipofectamine RNAiMAX Transfection Reagent (Life Technologies). Two days after transfections, cells were lysed for biochemical asssays or fixed for immunolabeling assays. UBR3 siRNAs (Sigma MISSION Predesigned siRNA) used in this paper: UBR3 siRNA791: GUUAGAAGGCGCUCUUACA; UBR3 siRNA793: GUACUUAAGAGAAGGCUAU; UBR3 siRNA795: CCGAAAUGUUGUUAGAUAU

## Co-immunoprecipitation and western blot

S2 cells were lysed 48 hr after transfection with plasmids in lysis buffer (Tris-HCl 25 mM, pH 7.5, NaCl 150 mM, EDTA 1 mM, NP-40 1%, Glycerol 5%, DTT 1 mM) plus complete protease inhibitor (Roche, Switzerland) for 30 min on ice, followed by centrifugation. The supernatant was then immunoprecipitated with agarose beads conjugated to antibodies recognizing different epitope tags, which had been previously equilibrated with lysis buffer, overnight at 4°C. The beads were then washed 3 times in washing buffer (Tris-HCl 10 mM, pH 7.5, NaCl 150 mM, EDTA 0.5 mM) before boiling in loading buffer. Western blotting was then performed with each sample. To purify MyoVIIa-GFP proteins from clone cells in the eye-antennal discs, tissues were homogenized in RIPA buffer (Tris-HCl 50 mM, pH 7.5, NaCl 150 mM, sodium deoxycholate 0.25%, NP-40 1%, SDS 0.1%). Immunoprecipitation was performed using the same conditions as above, except for the experiment shown in *Figure 5—figure supplement 1C*, in which MyoIIa was purified through immuno-precipitation in a denatured condition (*Bloom and Pagano, 2005*) to avoid pulling down interacting proteins. The following affinity beads were used for immunoprecipitation: Chromotek-GFP-Trap Agarose Beads (Allele Biotechnology, San Diego, CA), monoclonal anti-HA−agarose antibody (Sigma), anti-V5 agarose affinity gel (Sigma), monoclonal anti-MyoIIa (abcam ab55456, 1:100, used in *Figure 5J and K*). The antibodies used in western blot analysis are as following: rabbit anti-GFP (1:1000, Life Technologies), rabbit anti-MyoII (1:1000, a gift from Dr. Dan Kiehart, used in *Figure 3B* and *Figure 3—figure supplement 1G*) (*Kiehart and Feghali, 1986*), mouse anti-MyoVIIa (1:10, DSHB) 138–1, used in *Figure 5J*) (*Soni et al, 2005*), mouse FK1 (1:1000, Enzo, Farmingdale, NY), mouse FK2 (1:1000, Enzo), anti-HA (1:5000, Santa Cruz Biotechnology (Santa Cruz, CA), F7 or (1:1000, 16B12, Covance, Princeton, NJ), guinea pig anti-Ubr3 (1:1000) (*Zanet et al., 2015*), mouse anti-V5 (1:5000, Life Technologies), rabbit anti-MyoIIa (Gene Tex GTX113236, and 1:1000 for western blot in *Figure 5J*). The intensities of the bands in *Figure 3B* were quantified using Image J software.

## Quantification of detached scolopidia

For MARCM-mediated knockdown or mutant experiments, the percentage of detached scolopidia was calculated as the number of detached scolopidia in GFP clones / the total number of scolopidia in GFP clones. For *Cad99C57A* homozygous mutants, detached scolopidia in the each Johnston's organ were counted and the percentage was calculated by dividing the detached scolopidia number by 230, the average number of scolopidia per organ (*Kamikouchi et al., 2006*). Statistical calculations were computed using Prism 3.0 software.

## Electrophysiology

Electrophysiological recordings were performed with electrolytically sharpened tungsten electrodes inserted into the joint between the first and second antennal segments (recording electrode) and penetrating the head cuticle near the posterior orbital bristle (reference electrode), in response to near-field playback of computer-generated pulse song, as described in (*Eberl and Kernan, 2011*). The signals were subtracted and amplified with a differential amplifier and digitized at 13 kHz. Sound evoked potentials (SEPs) were measured as the max-min values in the averaged trace from 10 consecutive presentations of the pulse song, as described.

## RNA extraction and RT-PCR

Total RNA was isolated from ARPE-19 cells using Trizol (Life Technologies). Reverse transcription was performed using the Applied Biosystems High-Capacity cDNA Reverse Transcription Kit. RT-PCR was performed using iQ SYBR Green Supermix from BIO-RAD (Hercules, CA) and CFX96 Touch Real-Time PCR Detection System. Primers used for the RT-PCR were: UBR3-F (5′-TGGCTG TTCAAGGTTTCATAGG-3′) and UBR3-R (5′- GGTGCCACTGCTTAGTTTTACC-3′), GAPDH-F (5′-AA TCCCATCACCATCTTCCA-3′) and GAPDH-R (5′-TGGACTCCACGACGTACTCA-3′). RT-PCR was done with 3 PCR replicates for each biological sample, 3 biological replicates (3 independent biological samples in the same experiment) and was repeated twice (2 independent experiments).

## Acknowledgements

We thank the Bloomington *Drosophila* Stock Center, Vienna *Drosophila* Resource Center and Drosophila Genomics Resource Center for fly stocks and cDNA clones. We thank Dr. Cheng-Ting Chien, Dr. Yun Doo Chung, Dr. Christian Dahmann, Dr. Janghoo Lim and Dr. Fengwei Yu for sharing fly stocks, antibodies and constructs. We thank Hongling Pan and Yuchun He for injections to create transgenic flies and Hsin-I Jen for experimental help. The rat anti-elav monoclonal antibody (7E8A10, developed by G Rubin), mouse anti-Prospero monoclonal antibody (MR1A, developed by C Doe), anti-Repo monoclonal antibody (8D12, developed by C Goodman) and mouse anti-MyoVIIa antibody (138-1, developed by DJ Orten) were obtained from the Developmental Studies Hybridoma Bank, created by the NICHD of the NIH and maintained at The University of Iowa, Department of Biology, Iowa City, IA 52242. Confocal microscopy at BCM is supported by the Intellectual and Developmental Disabilities Research Center (NIH 5P30HD024064). This work was supported by NIH DC010987 (AK Groves), and the Howard Hughes Medical Institute (HJ Bellen). This work was also facilitated by the Iowa Center for Molecular Auditory Neuroscience, supported by NIH P30 grant DC010362 to Steven Green. HJ Bellen is a Howard Hughes Medical Institute Investigator.

## Additional information

### Competing interests

HJB: Reviewing editor, *eLife*. The other authors declare that no competing interests exist.

### Funding

| Funder | Grant reference number | Author |
| --- | --- | --- |
| National Institutes of Health | DC010362 | Daniel F Eberl |
| National Institutes of Health | DC010987 | Andrew K Groves |
| Howard Hughes Medical Institute | | Hugo J Bellen |

The funders had no role in study design, data collection and interpretation, or the decision to submit the work for publication.

### Author contributions

TL, Conception and design, Acquisition of data, Analysis and interpretation of data, Drafting or revising the article, Contributed unpublished essential data or reagents; NG, AKG, HJB, Conception and design, Analysis and interpretation of data, Drafting or revising the article; DFE, Conception and design, Acquisition of data, Analysis and interpretation of data, Drafting or revising the article; SNJ, DG, Conception and design, Drafting or revising the article, Contributed unpublished essential data or reagents; TC, Conception and design, Acquisition of data, Analysis and interpretation of data

### Author ORCIDs

Hugo J Bellen, http://orcid.org/0000-0001-5992-5989

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
