## [Decision Letter]

Thank you for submitting your article "The E3 ligase Ubr3 regulates Usher syndrome and *MYH9* disorder proteins in the auditory organs of *Drosophila* and mammals" for consideration by *eLife*. Your article has been reviewed by three peer reviewers, one of whom, K VijayRaghavan, is a member of our editorial board and also oversaw the process as Senior editor. Andrew Jarman (peer reviewer) has agreed to reveal his identity.

The reviewers have discussed the reviews with one another and the Reviewing Editor has drafted this decision to help you prepare a revised submission.

Summary:

This manuscript makes a significant advance in our knowledge of the proteins that underlie USH1 (Usher Syndrome type I), which function in mammalian auditory mechanotransduction. Strikingly and unexpectedly, convincing evidence is provided that these proteins are not only conserved in *Drosophila*, but also their interactions are important for the *Drosophila* auditory apparatus. The significance of this is that it establishes *Drosophila* as a model for future general understanding of USH1's role in mechanotransduction.

In addition, evidence is provided for a role for ubiquitination of *myoII* as a key step in regulating USH1 interactions in *Drosophila*. Although the roles of various E3 ligases in this is clearly not straightforward, the overall conclusion appears sound and this represents a useful mechanistic advance. Importantly, this ubiquitination appears to be conserved in mammals, although this is mostly on the basis of experiments in an ARPE-19 cell line.

Essential revisions:

One general important comment is that this potential conservation of mechanism (which strongly implies its importance) would be further substantially strengthened if the 'smoking gun' of ubiquitinated mammalian MyoII could be detected (e.g. by IP) in the cells or in the cochlea. However, the data as currently presented will clearly be a trigger for future studies in mammals.

In addition, we have some specific experimental suggestions to that can bolster conclusions made on the mechanisms of action. The authors should decide which can be done speedily to strengthen the conclusions of this study. While points 1 and 2 (below) are straightforward fly genetics if the directions suggested are embarked upon, we do appreciate that they may not be doable in the 8 weeks’ time we would like to see for resubmission (unless the authors have already embarked on these directions). The authors may want to consider other speedier experimental directions or make a case why the current data suffices. In other words, while we would like more data on this front, we are prepared to be convinced by other compelling experiments or arguments.

Specific points to be addressed.

1) The Ubr3 and SCF regulate indirectly the mono-ubiquitination of Myosin II and the interaction with Myosin VIIa. The involvement of E3 ligase activity of Ubr3 could be addressed more directly such as by making specific point mutations.

2) The effect of mono-ubiquitination on Myosin II does not appear strong in the assay used. One way to resolve this is to identify the ubiquitination site and test its effects.

3) Myosin II and Myosin VIIa do not seem colocalized while using mammalian cells, which is difficult to reconcile with the fly results. Also whether the dis-localization on stress fiber in UBR3 knockdown is due to the absence of stress fibers or due to mislocalization is not clear.

4) It is not clear whether human-mimetic mutations in Figure 4 are related to UBR3 regulation or Myosin VIIa interaction. Examining their mono-ubiquitination status and interaction could resolve this.

5) While including Cad99C and Sans to describe the phenotypes and interaction with Cul1, Myosin VIIa and UBR are nice, it is not clear how these components could be put together to delineate a significant mechanism? For example, are they substrates for ubiquitination by Ubr3 or SCF? This point could be addressed in the Discussion.

---

## [Author Response]

Essential revisions:

One general important comment is that this potential conservation of mechanism (which strongly implies its importance) would be further substantially strengthened if the 'smoking gun' of ubiquitinated mammalian MyoII could be detected (e.g. by IP) in the cells or in the cochlea. However, the data as currently presented will clearly be a trigger for future studies in mammals.

In response to this comment, we purified MyoIIa from ARPE-19 cells through immuno-precipitation and detected the mono-ubiquitinated form of MyoIIa using the FK2 antibody in ARPE-19 cells. This data is now shown in Figure 5.

*In addition, we have some specific experimental suggestions to that can bolster conclusions made on the mechanisms of action. The authors should decide which can be done speedily to strengthen the conclusions of this study. While points 1 and 2 (below) are straightforward fly genetics if the directions suggested are embarked upon, we do appreciate that they may not be doable in the 8 weeks’ time we would like to see for resubmission (unless the authors have already embarked on these directions). The authors may want to consider other speedier experimental directions or make a case why the current data suffices. In other words, while we would like more data on this front, we are prepared to be convinced by other compelling experiments or arguments.*

*Specific points to be addressed.*

*1) The Ubr3 and SCF regulate indirectly the mono-ubiquitination of Myosin II and the interaction with Myosin VIIa. The involvement of E3 ligase activity of Ubr3 could be addressed more directly such as by making specific point mutations.*

To address this comment, we generated transgenic flies with an E3 ligase dead form of Ubr3 to assess if this mutant version rescues the scolopidial detachment in *ubr3* mutant clones in the Johnston’s organ. This E3 ligase dead form has no E3 ligase activity (Li et al., 2016). We now show that this E3 dead form of Ubr3 fails to rescue the detachment phenotype in *ubr3* mutant cells while the wild type Ubr3 rescues the phenotype. This provides further evidence that the ligase activity is indeed necessary for its function in the auditory organ. These data are now shown in Figure 2—figure supplement 1.

*2) The effect of mono-ubiquitination on Myosin II does not appear strong in the assay used. One way to resolve this is to identify the ubiquitination site and test its effects.*

Although this is an excellent point, and we are planning to do these experiments, it will require a mass-spec experiment, a mutagenesis experiment, creating transgenes, and testing their phenotype to address this question. This is likely to take about 5-6 months. Hence, we cannot provide the answer to this question.

*3) Myosin II and Myosin VIIa do not seem colocalized while using mammalian cells, which is difficult to reconcile with the fly results. Also whether the dis-localization on stress fiber in UBR3 knockdown is due to the absence of stress fibers or due to mislocalization is not clear.*

We are confused by this comment as we show in Figure 5 that most MyoVIIa and MyoIIa co-localize in the stress fiber (arrows in Figure 5) and in cytosolic punta in the ARPE-19 cells (solid arrowhead in Figure 5). We agree that some MyoVIIa proteins do not co-localize with MyoIIa in the peri-nuclear region (empty arrowhead in Figure 5). To address whether the dis-localization of MyoIIa on stress fiber in UBR3 knockdown cells is due to the absence of stress fibers or due to mis-localization, we stained ARPE-19 cells transfected with UBR3 siRNA or treated with Bleb with phalloidin, which label the stress fibers. We show that the stress fibers are decreased in the UBR3 knock-down cells or in the cells treated with Bleb, suggesting that stress fiber formation is affected in these cells. The data is now shown in Figure 5—figure supplement 1.

*4) It is not clear whether human-mimetic mutations in Figure 4 are related to UBR3 regulation or Myosin VIIa interaction. Examining their mono-ubiquitination status and interaction could resolve this.*

To address these questions we first tried to express the mutant MyoII proteins in the proper cells and at the proper time, essentially mimicking the endogenous expression profile of MyoII. To this end we generated a *myoII*-T2AGal4 in which the coding sequence of Gal4 is inserted into an intron of *myoII*. This abolishes gene function but now permits Gal4 expression under control of the endogenous regulatory elements of the locus. Using this Gal4 driver we expressed wild type MyoII as well as the different MyoII variants. Expression of the wild MyoII did not cause any obvious defects and the animals were viable. However, expression of the MyoII pathogenic variants caused early larval lethality. This is consistent with a dominant toxic effect of MyoII variants. To bypass the lethality, we used *ey*-Gal4 to express the pathogenic variants of MyoII or wild type MyoII in eye-antennal discs and performed IP to assess their interactions with MyoVIIa. We found that MyoII^D1847K^ exhibited an increased interaction with MyoVIIa when compared to wild type MyoII. These data are shown in Figure 4—figure supplement 1. The data for the other three are either not changed or there is a decrease in their interaction. However, we were not able to purify enough MyoII protein using *ey*-Gal4 to detect the ubiquitination of MyoII (too little tissue). Hence, we cannot assess if the ubiquitination of MyoII variants are affected.

*5) While including Cad99C and Sans to describe the phenotypes and interaction with Cul1, Myosin VIIa and UBR are nice, it is not clear how these components could be put together to delineate a significant mechanism? For example, are they substrates for ubiquitination by Ubr3 or SCF? This point could be addressed in the Discussion.*

We added the following to the Discussion:

“It is interesting that Ubr3 and Cul1 can bind to a complex containing MyoVIIa, Cad99C and Sans in *Drosophila* S2 cells. It is possible that Ubr3, Cul1 and the unknown E3 ligase that mono-ubiquitinates MyoII all interact in a protein complex, and that the Usher proteins Cad99C or Sans may facilitate ubiquitination of MyoII or may be ubiquitinated themselves.”